# Responding to a Weeping Planet: Practical Theology as a Discipline Called by Crisis

**Mary Elizabeth Moore** 

School of Theology, Boston University, Boston, MA 02215, USA; memoore@bu.edu

**Abstract:** Practical theology is by nature a discipline of crisis, standing on the edge of reality and potential, what is and what can be. Crises can be gentle turning points, opportunities for radical transformation, or catastrophic moments in time. In the geological age of the Anthropocene, people face devastating planetary effects of human agency, which have created and escalated a climate crisis beyond the boundaries of imagination. Practical theology belongs at the epicenter of ecological crises, which have already produced harsh results, ecological despair, and a time-dated urgency for daring decisions and actions. Change is knocking at global doors—the necessity, foreboding, and hope for change. This article probes practical theology's role in change, giving primary attention to changes in practical wisdom (phronesis) and life practices. Methodologically, the article draws from ecological scholars and activists, philosophers and theologians, indigenous communities, and the earth itself, presenting descriptions and analyses of their shared wisdom across time, culture, and areas of expertise. From these sources, the study identifies challenges, practices, and alternate worldviews that can potentially reshape practical wisdom and climate action. In conclusion, this paper proposes life practices for climate justice: practices of attending, searching, imagining, and communal living and acting.

**Keywords:** practical theology; crisis; practical wisdom; ecological justice; life practices; climate change



## 1. Background and Methods

One day in the summer, I walked down the street in my Boston home, where smoke filled the sky and burned my eyes. The smoke had traveled here from Canada and the Western United States, where wildfires were storming across the land. Smoke and fire debris had etched the skies across North America, and the western fires continued to break records in terms of heat produced and land destroyed. A few days later, the Dixie Fire in Northern California was depositing smoke and ash in a continual flow within a 1000 mile radius of its flames. Weeks later, the western fires had further heightened the vulnerability of the atmosphere to extreme weather conditions, including raging rains and floods, and culminating yet again in mud slides and radical damage to soil, clean water, hillsides, forests, habitats for animals, and plants. As fires speed deforestation across the world, rising temperatures increase drought conditions, and dramatic increases of evaporation escalate catastrophic rainfalls, some of which are in the very regions where the heat has risen and forests and foliage have dramatically decreased, and some many miles away. Compound events continue to rise.

We in North America are facing ecological crisis on a massive scale, as are lands and peoples across the world. On our shared continent, we face crises of water and habitats as well as forests aflame, intensifying storms, and dramatic changes in temperatures and rain and snow falls. Consider the rivers that no longer provide habitat for salmon and other fish, the aquifers that have diminished or disappeared, the land that has been stripped of its nutrients in the quest for industrialization and corporate and individual profit, and the habitat networks that have been diminished and separated by industrial or human development so that animals and plants are less and less able to travel between habitats in

changing conditions. All these issues are interconnected both physically and socially, and are well documented in international scientific studies (IPCC—Intergovernmental Panel on Climate Change 2021; IPCC—Intergovernmental Panel on Climate Change 2022).

A striking example is the Keystone XL (KXL) Pipeline, an extension of Keystone that was stopped after more than 10 years of struggle; it would have run through five provinces and states and through tribal lands. The southern portion of that pipeline extension had already been completed, running through two new states, as it added to the existing pipeline that already ran through eleven provinces and states, carrying tar sands oil from the borealis forest of Alberta, Canada, for conversion into flowing oil in the United States. The KXL project, if completed, would have dramatically increased the rate of tar sands extraction, destruction of the boreal forest, rate of leakage and spills (higher than for other kinds of oil), destruction of farms and habitable land, and greenhouse gases and carbon released in processes of extraction and refining. In all these escalations, the pipeline threatened to escalate the poisoning of indigenous lands and peoples. These are the same people who have been affected by other industrial development projects that have dammed and diverted rivers, leading to water shortage in some areas and flooding in others. Politically, the people most affected—indigenous communities, farmers, ranchers, and others—are rarely included in decision-making before projects are formulated, or even before projects are set in motion by industries and governments. The dangers and disasters described here represent a complex of crises that date back to the arrival of European settlers, displacement of indigenous peoples, and hundreds of decisions that benefited industry and harmed the people who live on this land, while wreaking havoc on the land itself. Innumerable activists protested the KXL crisis with words and actions and helped stop the extension from becoming a reality (Gilio-Whitaker 2019, pp. 10–24; McKibben 2013, pp. 17–44).

What is the nature of these crises, and how do they relate to practical theology? Methodologically, I will focus on the nature of crisis and then turn to the practical wisdom (phronesis) of ecological scholars and activists, theologians and philosophers, and indigenous communities as they approach earth wisdom and earth caring in their unique ways. By analyzing these sources, I will portray their shared wisdom across time, culture, and areas of expertise. Curiously, their various approaches all include some version of reflection on practices for the sake of future practice, disclosing a resonance with the work of practical theology. On the other hand, indigenous people and many of the activists reflect on practices of the entire natural world, not just human beings, thus challenging the dominant anthropomorphic focus of practical theology. The entire study illumines how diverse scholars and activists contribute to practical wisdom, or phronesis, through reflection on their unique experiences, and how they contribute to human ways of being in and with the world. In particular, the study identifies challenges, practices, and alternate worldviews that can potentially reshape practical wisdom and climate action, and it concludes with proposals for practice toward climate justice.

## 2. Crises

Crises are turning points or moments when forces meet, facing people and other beings with critical decisions that will shape the future. They are often associated with dramatic or disastrous moments, but the term is also broad. Crises exist in daily lives when people live detached from God's creation and from communion with other people and trees and soil. Crises appear when people face choices whether to continue their customary life patterns or consider alternative ways of relating with the earth, with other creatures, and with themselves. These daily crises (often unrecognized) are connected with catastrophic ones. The catastrophic crisis of climate change was set in motion by countless decisions made in everyday life over countless years to suck life from the planet for human gain and without regard for the life of the planet and all that lives on it. Likewise, the catastrophic plight of indigenous peoples in North America did not emerge in one traumatic moment, but in millions of traumatizing acts of war and murder; kidnapping and enslavement;

colonialization; displacement; broken treaties; and destruction of lands rich with fish, clean water, plants, soil, and game.

Crisis moments often go unnoticed without conscious reflection, especially when they are part of prevailing life patterns and worldviews that are taken for granted; yet people make small and large decisions to act in certain ways, which affect both the immediate and long future. Consider how theories of individual autonomy and individual well-being, taken for granted in most Western Christian cultures, have conflicted with indigenous theories of relationality and communal–ecological well-being, leading to the overtaking of land for individual gain or the economic gain of industrialists. These actions can be cataclysmic over time, creating and reinforcing patterns of injustice that destroy peoples, cultures, and the earth. The actions often appear innocuous at first, but the narratives of European settlements in North America reveal how settler communities, individuals, churches, and state/provincial and federal governments have contributed to the amassing of injustice toward indigenous peoples and toward the earth.

Beneath these actions are theologies or worldviews that generate and reinforce the actions. The interplay of practices and worldviews reveals the dynamic of what Don Browning describes as theory-laden practice and practice-laden theory, as well as "tradition-saturated ideals and practices" ([Browning 1991](), pp. 5–7, 10–12, 45–49). For him, any ethical or theological question needs to be explored with these complex relationships in mind: theory, practice, and tradition. I add poesis to this list to acknowledge the role of creativity and beauty.

Whether far-reaching or small and local, crises were turning points for European settlers in North America as they made daily judgments and decisions: how best to live in this land; whether and where to clear forests for farming; how to keep their people "safe" from peoples they feared or considered less than themselves; how to establish Christianity in this land; or how to ensure control over land, water, and peoples different from themselves. The decision points took many forms. They were *habitual*, continuing engrained patterns; *anticipatory*, seeking to build or protect a future; and/or *reactive* to immediate situations. All of these together have brought the world to a point of catastrophe.

Practical theology is by nature a discipline of crisis, standing on the edge of reality and potential, what is and what can be. The multiple meanings of crisis are all relevant in this geological age of the Anthropocene in which human agency is the dominant determinant of the fate of the climate and earth. Crisis can be understood as a troubling situation, a moment for decision-making, and/or a moment in which a bad situation is likely to become markedly worse or better. Within macro-crises are also micro-occasions or events described in process-relational theologies in which the objective past and habits of being are met with God's initial aim and the potential of a new decision. An emerging event, through its subjective aim, sets a direction and a new occasion is formed, only to be followed by billions of others. This micro process also takes place in more complex organisms as a fox or forest or human being, as they carry their respective pasts, relationships, and accumulated wisdom into a new moment of time, making a new decision for this moment. The decisions may be repetitive of the immediate and dominant streams of the past, or they may move in novel directions. I propose that we recognize all these decision moments as crises, each representing change (however small or large) and each revealing the necessity, foreboding, and hope for change.

The challenge is for practical theologians to attend even more intentionally to the dynamics of change, building on current movements. For example, practical theologians regularly respond to habituated practices with *proactive responses*, encouraging people to examine their habitual patterns for the sake of comprehending, critiquing, and reshaping or reinforcing them. Consider the work of public practical theologians, cultural analysts, pastoral counselors and caregivers, educators, and homileticians. Climate crisis underscores the need for these same theologians to reflect on engrained habits that have contributed to ecological crises, and the habits that are needed to prevent further destruction in the future. Closely related, practical theologians also engage crises with *anticipatory responses*,

analyzing historical and contemporary trajectories of practice (e.g., practices of destructive human relationships or ecological neglect) and anticipating alternative futures. Finally, practical theologians often offer *reactive responses*, addressing threats of disaster, analyzing actions and underlying causes, and posing immediate responses to large-scale threats. In this Anthropocene age, all three responses are needed, as they flow in and out of one another and often take place simultaneously.

Practical theologians have largely focused on crises in the individual and community lives of human beings, but the time has come when that focus needs to be amplified with sustained attention to *all* the natural world and to ecological crises, seeking to protect and regenerate a threatened planet. What is tragically clear in the global climate crisis is that reactive responses are urgent, but the necessity for proactive and anticipatory responses is clear as well. Urgency cannot turn people away from the important work of proactivity and anticipatory action, as it often does. In all this work, practical theologians have an important role to play with their specialized attention to theory-laden practice, practice-laden theory, and tradition-saturated ideals and practices. To play that role requires that we attend to the relationship of practical wisdom and life practices.

### 3. Practical Wisdom: Reshaping Theology, Worldviews, and Relational Patterns

One of the profound challenges for addressing climate crises is the inadequacy of dominant theologies and world views, intertwined with inadequate relational patterns, as in colonial structures, patterns of dominance of some people over others, emphases on individualism and progress, and the dominance of humans over the land and sea. A prime example is the anthropocentricism of most Christian theologies, focusing on the superiority of humans over all other life forms, the salvation of human beings without reference to other beings, and the accompanying idea that the nonhuman natural world is a resource for humans to use and enjoy, or to tend as an inferior "other." Such views are often intensified in their ecological effects by elitist social structures in which individuals, social groups, and companies with the most economic and political power determine the "uses" of the eco-system and the limits to its protection. In anthropocentric, hierarchical worldviews, the flourishing of the planet is set aside, and the world's complex patterns of mutuality and interdependence go unrecognized. Such views lead to environmental strategies based on what appears best for people (or for people with power) without regard to the value of other beings, like animals, plants, and bodies of water.

What is needed is a deeper engagement with practical wisdom (or *phronesis*), or *deeper understandings of the world as formed in the process of living, relating, and reflecting over time*. Phronesis is a word with a complex of meanings. Different translations of Aristotle's *phronesis* highlight different emphases, most prominently practical wisdom, practical reason, practical virtue, prudence, or moral understanding. Each of these has unique nuances, but each represents a large view of the world. In my effort to capture the full-bodied, guiding force of phronesis, I use the term *practical wisdom*, which is shaped by relationships, practices, worldviews, and moral understandings. It thus includes a capacity to discern common good or ethical futures, and to discern pathways toward goodness. When practical theologians attend to crises of climate change, the turn to practical wisdom is critical—attending to practice and reflection on practice, to moral issues in the practices, and to the potential of alternative futures and actions that contribute to those futures.

The turn to *phronesis* is not new to practical theology. We see it in Don Browning, who emphasizes practical reasoning in the tradition of Aristotle and many later philosophers, recognizing the interpenetrating relationship between philosophies, theologies, and worldviews and the practices of human beings. Bernard Lee and Thomas Groome also accent phronesis, but with emphases on future good and ethical action toward that good. I stand more in the tradition of Lee and Groome, but I also accent the fullness of knowing through all the senses and with all one's being, a view more closely aligned with Jennifer Ayres (Ayres 2013, pp. 37–38). Such holistic knowing is vital as people question and transform

theologies and worldviews to reflect more fully on God–world relationships and evoke visions of ecological justice.

My working definition of practical wisdom is the embodied, accumulating knowledge and ethical insight that arise from human and creaturely experience of the world and the numinous. Note that I include the experience of both humans and other creatures, thus recognizing that trees and grasses and animals and people are knowing creatures, albeit in diverse ways, and they have capacities to communicate with one another. Further, all beings have multiple ways of relating with the earth—through senses and sensors, fact-gathering, intuitive or instinctive impulses, esthetic experience, and analysis. Both human and earth knowledge accumulate over time, and values are inherent in the knowing, e.g., the value of preserving life. For humans, practical wisdom is never complete, and it is never without ethical influence. It is a way of knowing and being in the world, and is critically important if we are to understand and respond well to the climate crisis.

### 3.1. Challenging Practical Wisdom

The dominant forms of practical wisdom neglect much of the world, often justifying social, economic, and ethnic stratifications and human abuses of land and seas for the sake of monetary or techno-industrial progress. Ecologists have long recognized the need to question dominant world views and theologies, seeking to replace individualistic views with communal ones; accents on progress with accents on well-being; anthropocentric views with ecocentric or cosmocentric ones; technological solutions with holistic ecological ones; focus on economic expansion with focus on sustaining and regenerating the ecology; political competition with communal collaboration; and hierarchies of power with deep listening to all beings of creation. I will focus here on the issues and insights posed by some of the most influential ecologists of the past 80 years, recognizing that equally important challenges and alternatives are offered by indigenous peoples, to whom I turn in the next section.

Aldo Leopold was an early pioneer in the modern ecological movement, especially with the posthumous publication of his seminal *A Sand County Almanac* (1949). He was far beyond his time in his understanding of the urgent need for compassion and justice for the whole natural world, including human beings, who were both a beneficiary and a danger to the rest of the natural world. Leopold's land ethic was a guide for human understanding and responsibility, though seriously marred by a lack of racial sensitivity and racist comments. He saw a contrast between conservation and Abrahamic concepts of land, which he identified with commodification. In his foreword, he argued, "We abuse land because we regard it as a commodity belonging to us. When we see land as a community to which we belong, we may begin to use it with love and respect" (Leopold [1949] 2013, pp. 16–17). *Herein lies a major challenge to practical wisdom—to reject worldviews of commodification and to understand the land as community.* An additional challenge has to be addressed to Leopold himself, namely *the challenge to question hierarchies of value, including those that value some peoples over others.*

Rachel Carson carried a different, but complementary, message. She was a marine biologist, ecologist, and science writer, who had a gift for communication in scientifically accurate, poetic prose, thus catching the attention of a broad public. In her earliest book, *Under the Sea Wind*, she told stories of sea animals—birds, fish, turtles, and other creatures—as they lived their daily lives (Carson [1941] 2007). This is a series of narratives that invite readers into the actual worlds of these creatures and their waters. Carson challenges the practical wisdom of viewing the world from human perspectives by narrating an ecology of life among animals related to the sea and one another. In sum, *she challenges dominant forms of practical wisdom by decentering human experience.* Carson ([1962] 2002) offers another challenge in her most famous book, *Silent Spring*, in which she explains the effects of chemicals and pesticides on humans and other living beings, focusing particularly on DDT and its uses in agriculture. Here *her challenge is twofold, revealing the permeability of humans to chemicals, to poisons and contaminants outside themselves, and revealing the enormous damage*

*that can be done to people, plants, and land by chemical pesticides*. This particular work won widespread attention and eventually led to the elimination of DDT and other pesticides from U.S. agricultural use, but sadly not from exports.

Others continued the accents of Leopold and Carson. One of the people who explicitly raised issues of worldviews and practices was the educator Chet Bowers. He sought to connect ecojustice with educational practices in childhood, youth, and higher education (Bowers 1995, 1997, 2001, 2016). From 1974 to 2016, Bowers addressed *worldview challenges, explicitly worldviews grounded in individualism, progress, economic–political power, and human dominance over the earth*. He argued for a "deep cultural" movement, critiquing education for reproducing the very cultures that are destroying the earth by perpetuating the cultural myths that magnify ecological crises. Bowers decried simple answers, even the practice of blaming capitalism; he made a case that *all* assumptions need to be questioned (Edmundson 2017). As a fierce advocate for changes in worldview, he drew especially on indigenous traditions and values to discern and encourage cultural alternatives.

Many other advocates for ecological justice and flourishing have offered ecological alternatives and reshaped worldviews, drawing upon Christianity, Hinduism, Buddhism, Confucianism, indigenous traditions, and others (Grim and Tucker 2014; Tucker 2003; Vaughan-Lee 2016) and upon particular theological traditions such as African American (Baker-Fletcher 1998); Lutheran and ecumenical (Rasmussen 2015; Chicka 2019; Santmire 2020); and process–relational (Cobb 2020; Cobb and Castuera 2015; Birch and Cobb 1981). In all these and more, people have questioned and proposed alternatives for the development of practical wisdom, echoing the challenges named above.

Some ecological authors of the past 40 years have also highlighted the close relationship between environmental destruction and the ideological or physical crushing of women and the poor. Women have frequently raised issues of the mutually reinforcing oppression of the earth, women, and marginalized peoples (Johnson and Wilkinson 2020; Shiva 2015, 2016a, 2016b; Ruether 1995, 1996, 2006; Gebara 1999, 2003; Halkes 1989). Some compilations have brought together voices of women across the world (Ruether 2006) and activist women in diverse cultures and roles (Johnson and Wilkinson 2020). In addition to analyzing gender, gender identities, and environmental perspectives, these women often identify with and speak strongly on behalf of the poor, as do liberation theologians across the world (Boff [1997] 2000). The cries of the earth and the cries of the poor are intertwined. *The challenge is to explore ecological crises from the perspective of multiple religious and cultural traditions and multiple oppressions.*

I have portrayed some of the early work associated with the ecological justice movement to reveal the wide range of concerns and ideas from the 1940s to the present. What is still lacking is the hard work of listening deeply to one another and the earth itself, and the close questioning and reshaping of practical wisdom. What is also lacking is attention by practical theologians to issues of ecological justice and the climate crisis in particular. Pamela McCarroll has diagnosed the situation well, identifying the anthropocentrism of methods and goals in practical theology and in Western theologies more generally, together with the spreading of eco-anxiety that can stimulate defensiveness and thwart action (McCarroll 2020). HyeRan Kim-Cragg's critique is similar, with an emphasis also on the destructive forces of colonialism (Kim-Cragg 2018), as highlighted above by Gilio-Whitaker and others. While practical theologians have made a case for the field's engagement with public theology (Graham 2013; Forrester 2004) and social witness (Ayres 2019b), direct engagement with ecological injustice and crises is limited to fewer works (such as (Ayres 2013, 2019a, 2019b; McCarroll 2020; Kim-Cragg 2018)), even in the face of climate catastrophe. *The challenge now is to study both human and nonhuman practices—practices of trees and rivers and antelope—and to assess and reshape practical wisdom accordingly, so that daily life, community activism, and political action will be transformed.*

*3.2. Stretching Practical Wisdom*

To live toward a new future, we need to do more than analyze challenges; we need to stretch the largely white Western inheritance of practical wisdom. The previous section offered an arc of ecological concerns from the 1940s to the present, drawing largely on North American ecological leaders, mostly white. By itself, this approach is flawed. Gilio-Whitaker (2019) critiques the crediting of Rachel Carson's *Silent Spring* in 1962 and the first Earth Day in 1970 as the origins of a modern ecological movement, offering a more complex perspective. She especially decries the nineteenth century naturalists, such as John Muir, whose environmental perspectives were grounded in human superiority and colonial views of indigenous people. As much as they contributed, these hierarchy-oriented naturalists left a heritage that continues today in colonial approaches to the earth and to indigenous people. She encourages people to seek a fuller history (pp. 106–110). Thus, I turn primarily to North American indigenous authors in this section as they point to wider wisdom of the earth. In so doing, I acknowledge that other voices also need to be discussed in another work, voices from Africa, Latin America, Asia, Europe, and the Pacific. This too is urgent, as argued by Vanessa Nakate, a young Ugandan activist who decries the dismissal of African voices and points toward a "bigger picture" (Nakate 2021).

Dina Gilio-Whitaker lays much of the ecological crisis at the foot of colonial modes of theorizing and structuring the world, documenting case after case in which the domination of indigenous people and the destruction of ecosystems went hand in hand. She offers a sharp critique, supported by details of cases, laws, history, and activist efforts. She also describes traditions that undergird much indigenous life, including "a philosophical paradigm very different from that of dominant Western society," a paradigm that is often ignored in legal decisions and practices of appropriation (p. 157). She elaborates on the connections that indigenous people have with their homelands—the lack of "separation between people and land, between people and other life forms, or between people and their ancient ancestors whose bones are infused in the land they inhabit" (p. 157.). Others have also accented the profound connections indigenous people have with place and land, such as Vine Deloria, Jr., and George Tinker.

How can people in non-indigenous cultures value and learn from indigenous peoples without abusing and distorting their sacred traditions? The challenges of those traditions to most Western worldviews are so vital that we need to find ways to learn without objectifying and to collaborate without dominating or colonizing. Fortunately, some authors point the way, even as they warn about the dangers noted above. Sherri Mitchell, of the Penobscot people of Maine and the Maritimes, is an advocate of people's learning from one another and seeking harmony, even as she narrates massacres, Indian school tragedies, displacements, destruction of land, and the trauma of her people. She recognizes that the United States "was founded on genocide and slavery" (Mitchell 2018, p. 69), yet she appeals to the potential for healing. Her starting point is to recognize that humanity is at "a teetering point of choice that will determine the future of all life" (p. 22). Mitchell's vision is to weave Penobscot teachings with other views to harmonize and align "with common purpose," a vision of oneness without sameness (p. 23). To that end, she shares the sacred teachings of her people and invites readers to listen to and respect all living things (p. 28). Writing with a sense of urgency, she recognizes that the human family is on "the precipice of an evolutionary leap, one that requires us to transcend our differences and integrate into a more harmonized way of being" (p. 40).

Mitchell's gift is sharing the history and teachings of her people, paired with a vision of a world that can be. Her approach suggests how people might learn from one another and from all living things as they join in stretching practical wisdom. Robin Wall Kimmerer (2013) builds in a similar direction in *Braiding Sweetgrass*. She describes the worldviews of her Potawatomi people, a Native nation originating between Lake Michigan and Lake Huron (later dispersed) and akin to other tribes and first nations of the Great Lakes region in worldview, language, and life practices. Kimmerer herself lives and teaches in upstate New York, where she explores and lives with forests and family. In her writing, she

narrates the life of plants and her life among them, drawing from her knowledge as an environmental and forest biologist and the lifeways and traditions of her people. Like Mitchell, she understands her writing as an offering and an invitation for others to learn from the wisdom of her people and the life of plants.

Kimmerer highlights details that I identify as practical wisdom. She describes the wisdom of pecan trees that have airborne communication systems (via pheromones) to warn other trees against pests and connections, which in turn produce defensive chemicals. Pecan trees are also connected underground via fungi and fungal strands that, based on current evidence, gather the excess carbohydrates from some trees to share with others, thus spreading the bounty and enabling the whole pecan grove to bear pecans at the same time, rather than creating a divide between bearing and non-bearing trees. She sees the communication systems as the ways that trees talk with one another as her ancestors taught her to believe (pp. 11–21, esp. 19–21).

Kimmerer also describes the gifts of wild strawberries and black ash, living interdependently with the Potawatomi people, who gratefully receive nourishment from the strawberries and basket makings from the black ash, while other animals also receive from them and while people care for the plants and trees by careful selection and pruning (pp. 22–32, 141–55). Kimmerer further describes what she learns from maple trees and grasses and how they live mutually with other life forms in their habitats (pp. 63–71, 156–166). For the Potawatomi, primal values are grounded in gratitude and the aliveness of creation, which inspire practices of mutuality and reciprocity for the sake of the living whole.

Kimmerer further stretches practical wisdom as she describes the living nature of her native language, learned as an adult. Baffled by the complexity of words and grammatical structures, she experienced an epiphany when her sister sent her a box of word tiles and an Ojibwe dictionary, given the close relationship between her language and that of Ojibwe people. Kimmerer was ready to quit when she discovered an unusual word, "to be a Saturday" (p. 66). How could a noun be expressed in verb form? She reacted: "I grabbed the dictionary and flipped more pages and all kinds of things seemed to be verbs: 'to be a hill,' 'to be red,' 'to be a long sandy stretch of beach,' and then my finger rested on *wiikwegamaa*: 'to be a bay,' 'Ridiculous!'" Then, in a sudden moment of realization, she could smell and see and hear the bay. The moment was an epiphany: "A bay is a noun only if water is *dead*. When *bay* is a noun, it is defined by humans, trapped between its shores and contained by the word. But the verb *wiikwegamaa*—to *be* a bay—releases the water from bondage and lets it live" (p. 66.). The practical wisdom conveyed in these language forms communicates that everything is alive. Kimmerer calls this "the language of animacy," and the animacy extends to rocks and mountains, fire and places, and medicines and songs (p. 67). I have shared Kimmerer's discovery to reveal the nature of practical wisdom, embodied in language and in the relationships of pecans, strawberries, maple trees, and black ash with their habitats and the plants, people, and other animals that dwell therein.

Kimmerer's discoveries are not unique to her or her people. The Hopi language, while different from the Potawatomi, is also marked by relationships and movement, as discovered long ago by linguist Benjamin Lee Whorf (1956). He noted that the Hopi word to describe a wave in the ocean is not an isolated "wave"; rather, it appears in multiple verb forms that indicate a range of movement, such as sloshing, "kicking up a sea," or undulating. The words describe movements of the water in relationship with the sea, rather than isolating the wave from the movement in which it flows (pp. 52–53). Whorf explains that the Hopi preference for verbs contrasts with the English preference for nouns; thus, the Hopi language "perpetually turns our propositions about *things* into propositions about *events*" (p. 63). Space and time are merged, as are subjects and predicates.

Whorf's analysis of the complex Hopi language is more nuanced than my summary, but one can see the movement nature of the language, which echoes Kimmerer's description of movement in Potawatomi and Ojibwe languages. Whorf also observes connections between language and worldviews; "Most metaphysical words in Hopi are verbs, not

nouns as in European languages" (p. 61). Language has the power to shape and be shaped by worldviews; thus, people do not experience time and matter in the same ways because their experiences are mediated through diverse languages (p. 158). Language affects what people experience, and their experiences help shape language.

I have shared in some detail the teachings of indigenous peoples, pecan trees and wild strawberries, and diverse languages, all stretching the limits of practical wisdom in North America. If we are to respond well to a weeping planet, we need to face the limitations of the assumed wisdom that modern/postmodern people have inherited within a techno-industrialized, progress-seeking, individual-focused, capital-oriented society. We need to stretch practical wisdom in order to learn from many ancestors (including our own), many living beings, many language worlds, and the spiritual experiences and instructions of peoples living closely with the earth.

*3.3. Reshaping Practical Wisdom*

Practical wisdom is not sufficient if we hold tightly to dominant views and relational patterns, which keep us inside the confines of prevailing assumptions. What is needed first is *to face the limits of dominant views and patterns, and to stretch our own ways of thinking and relating* by engaging deeply and respectfully with the assumptive realities of diverse peoples, including ourselves. Second, we need *to engage with the traditions, texts, and practices of our own religious communities*, as exemplified by many who have explored questions about God, Christology, creation, and planetary well-being in Christian tradition (Copeland 2020a; Keller 2017; McFague 2008, 2021). These authors and others engage in the process of reshaping practical wisdom in dialogue with the traditions of Christianity. Another approach is generating new work on biblical texts. One representative of that work is Rebecca Copeland, who approaches texts with ecomimetic interpretation in which she invites readers to identify with the nonhuman characters in a text, such as the characters of water, birds, leaves (Copeland 2020b, 2020c, 2021). Such an approach decenters human-centered interpretations, and it also opens fresh questions and insights. Third, we need *to touch–see–listen–taste–smell the living earth*, receiving its wisdom and relating with mutuality. Such engagement contributes to reverence for wisdom itself, for ourselves and others, for the earth, and for the numinous.

These approaches magnify the potential for stretching and reshaping practical wisdom as we engage with ourselves, with the wisdom of diverse peoples and traditions, with the trees and seas, and with the sacred. In the next section we turn to practices that can enhance our human ability to critique and reshape practical wisdom, at the same time reshaping ethical values and relationships so we might respond to a weeping planet.

## 4. Life Practices

In the face of crisis, practices are critical, intertwined with the practical wisdom that shapes and is reshaped by practice. The significance of practice is clear in Rachel Carson's close observation of sea creatures, and Robin Wall Kimmerer's existential and scientific engagement with wild strawberries and pecan trees, and her study of the Potawatomi language. Here I identify four practices that are important to reshaping and deepening practical wisdom for ecological justice: attending, searching, imagining, and communal living and acting. To ground the proposals, I draw especially on the wisdom of activist scholars and scholar activists.

*4.1. Attending*

Virtually all writing on ecological justice begins with attending—attending to the gifts of the earth; the lives of plant and animal creatures and water, rocks, and soil; and the tragedies of land and habitat destruction, carbon escalation, heavy burdens on poor and marginalized communities, and global warming. Living in reverent relation with the ecological community is an important starting point for reshaping theologies, worldviews,

and practices. Reverence begins with attending: being present to, learning from, and caring with and for creation.

Diana Ventura describes the central method of practical theology as "prayerful attentiveness for human flourishing" (Ventura 2021), a definition rich in possibility. I propose stretching her definition to "prayerful attentiveness for cosmic flourishing." The term "prayerful" suggests a posture of openness to all that the world has to communicate, including the sacred endowment of every being. "Attentiveness" suggests that people open all their senses so they may experience the fullness of the earth in living detail. The aim of cosmic flourishing embraces and cares for the whole of creation. Following this path, practical theologians would attend to shrinking wetlands, communities suffering from food insecurity, and trees and forests. Their attending would not only include the damage and destruction thrust on these communities, but also the lifegiving practices of the wetlands, communities, and trees themselves, practices that reveal their beauty, strength, and potential. Others have already paved this pathway in works such as *The Hidden Life of Trees* (Wohlleben 2016) and *The Songs of Trees* (Haskell 2018). To be attentive is to walk among the trees and wetlands, opening to their wisdom, and to engage in interdisciplinary study so foresters like Wohlleben and biologists like Haskell and Kimmerer can teach us and awaken our senses.

Attending requires that we use all our senses beyond seeing and hearing, which often function as primary in Western societies. I advocate, with Richard Kearney, the practice of touch, which he calls our "most vital sense" (Kearney 2021). If we are to attend to creatures of the natural world, we need to touch and smell and taste as well as see and hear. All the senses reveal uniquely, and each can be employed as appropriate to the subject of our attending. Touching and smelling a leaf or the soil of a farm reveal far more than sight alone.

Attending also requires that we observe communities and systems, as Leopold, Bowers, Boff, Gilio-Whitaker, and Kim-Cragg have done. Separating land from people, one creature from others, a wave from an ocean, a tree from a forest will distort our knowing. Relationality is vital to understand if we are to expand practical wisdom. Climate scientist Katharine Hayhoe has discovered that the best way to communicate climate change with other people is to point out the tangible effects of climate on what matters most to them. She does not try to convince people to elevate the priority of climate change, but to recognize how climate change affects the issues they already hold as top priorities: health, families, jobs, the economy, community well-being, and the well-being of persons on the margins (Hayhoe 2020, pp. 136–37). Hayhoe is doing more here than offering practical advice; she is underscoring the importance of attending to communities and systems—to wholes. Attending to systemic relationships can uncover surprises, as when ecologist Jane Zelikova traces the relationship of ants with the seed dispersal of trees and climate change (Zelikova 2020, pp. 335–36). Attending involves observation and reflection on both details and wholes in the ecosystem.

Practical theologian Jennifer Ayres has offered a good example of attending to details and wholes in her work on food (Ayres 2013), which narrates particular stories and events as well as food systems, attending to them directly and through the work of others. Attending is a pathway to better understanding and action, deepening our practical wisdom and living practices.

### *4.2. Searching*

A second practice arises from the first, namely searching. Knowing and relating with the earth requires an attitude of openness, curiosity, and humility. Just as Kimmerer (2013) was eager to understand the plants in the pond behind her home, so Mitchell (2018) and Gilio-Whitaker (2019) were eager to understand the relationship between colonialism and ecological destruction. So, also, was Carson ([1962] 2002) eager to learn the effects of chemical pesticides on human, animal, and plant life. In all these cases, the authors searched to increase their understanding and guide their action, whether the action was

cleaning a pond, political advocacy, or the elimination of pesticides in agriculture. These author-activists shed light on the future work of practical theologians: (1) searching the ecosystems in our home contexts to inspire awe and inform our understanding and action; (2) searching the practices of congregations and people of faith; and (3) searching sacred texts, ritual patterns, and religious structures and practices to reflect on the teachings and values therein, and to critique and reshape them when we find them lacking.

Searching is a familiar theme in practical theology, one that is clearly important to a discipline rich in research and effective in shaping actions for the common good. Wanda Stahl, for example, has engaged in a long-term study of Wild Churches to discover their congregational practices and embedded theologies and values (Stahl 2021). The Wild Churches regularly gather in community in their local habitat (often with pets), and they reshape practical wisdom as they do. Stahl has discovered that their theologies often change over time, moving toward pan*en*theism and an accent on salvation for the whole planet.

Searching is a form of opening ourselves to that which we have only glimpsed, have never encountered, or have been avoiding. It is also a form of seeking reversals or asking hard questions (Moore 2004, chp. 5; 2021). The searching process can fill gaps in a community's knowing, or present major challenges as new discoveries, radical questions, and alternative futures unfold. It is not purely intellectual work, however, but quiet, meditative work as well. In theistic contexts, searching includes spiritual opening and discernment of God's movements, as well as complex deliberations on God and the world, and analysis of climate change and other crises. Searching is an endless practice, but it enriches the lives of searchers and the communities in which they live.

### 4.3. Imagining

Responding to crises also requires imagination, the capacity to discern opportunities and alternate futures. Imagination is grounded in the world that people know through life encounters and study. At the same time, imagination transcends the "known" world and soars into a world of possibility, offering a potential antidote to human despair and passivity, anger and blame, and life-defying ideas about God and the earth. Consider the ways ecological despair permeates the lives of individuals and communities, perpetuating a sense that no action can ever make a difference. Consider the patterns of blame that block people from considering new information and new possibilities. Consider the sense of hopelessness that is reinforced by theologies centered on original sin or on popular beliefs that humans are autonomous, self-serving beings.

One practice vital to imagination is storytelling. Kendra Pierre-Louis has focused on the problems and possibilities of narratives in telling ecological stories and spinning tales of new possibilities. She argues that some stories create problems for ecological responses, actively "hurting us" (Pierre-Louis 2020, p. 175). She offers an alternate story of the Wakanda, a fictional, roughly historical country in Africa that uses technology to maintain ecological health. She encourages readers to search for and create stories that narrate climate change as an opportunity (p. 179), seeking new and ancient stories; stories in sacred texts and traditions; and stories in our daily lives. The possibilities for stirring imagination through stories are boundless.

### 4.4. Communal Living and Acting

If we are to respond to the climate crisis with more full-bodied practical wisdom and robust action, we need to begin with community, embracing historical, contemporary, and ecological communities. Individualism is widely credited as a major factor in climate crisis and ecological destruction. The practical wisdom sought in this essay is accessible only through community, particular and global, human and nonhuman. When I wrote *Ministering with the Earth* (Moore 1998), I already knew that hope for the earth depended on an interdependent, intersubjective relationship between the human family and the rest of creation, hence ministering *with* the earth. I have come to see far more complexity in that

relationship and the urgency of radically reshaping practical wisdom and relationships if we are to practice "with." Responding to the climate crisis requires far-reaching changes in the ways we relate to and conceptualize the world, ways we live and act in community. Where do we begin?

I propose three paths of practice for communal living and acting. First is to remember the communities from which we come, the peoples and lands that have made us who we are and depend on us for their well-being. The memories may be sweet or harsh or bland, but they are part of the universe with which we are related. Many of the authors in this article share memories of people and lands that have formed them (Mitchell 2018; Kimmerer 2013), recognizing the formative and transformative power of memory. Memory could be included much more actively in practical theological research and action, searching the memories revealed in communities (human and nonhuman), habitats, religious practices, and literature and folklore.

If we are to respond to a weeping planet, a second path is to *draw upon the legacies of people and waters and lands to guide environmental action*. Those legacies will be mixed in their values, but critical awareness of multiple legacies is vital if we are to uncover fresh perspectives on climate protection and regeneration in a time of extreme urgency. Many of the human legacies will be offered by peoples whose ancestries or ways of living in the world are non-dominant. Tara Houska is one who awakened me when she highlighted the difference between life with her people (Zhaabowekwe, Couchiching First Nation) and the practices of environmental protection in corporate boardrooms, in which metrics and campaign outcomes dominate (Houska 2020, pp. 257–59). How might ecological leaders and practical theologians reshape environmental action and climate crisis if we reshaped standard boardroom procedure to account for the legacies of people who live close to the earth? Might board members spend more time walking the land, observing the life patterns of plants and animals, listening to people's stories, observing the geological and climatological patterns, and seeking truly radical approaches to the climate crisis?

Many people have drawn their ecological inspiration from the legacies of human and earth communities. Rachel Carson ([1962] 2002) lived closely with the legacies of oceans and farmlands as she reshaped scientific communication and farming practices. Heather Toney (2020) turned to her African American community and the eco-system to guide her in leading environmental organizations and serving as Mayor of Greenville, Mississippi. Her community legacy fed her, and she laments that communities are usually ignored in facing ecological issues, even though they are "vital for identifying real solutions" (p. 102).

A third path of practice is to *create new experiments and patterns of living in and with communities, local and global*. Practical theologians focus on many areas of life, so the potential actions are many. Leah Penniman (2018) draws upon the wisdom of her African American ancestors to create Soul Fire Farm and to practice farming in a way that is sustainable and life-giving today, guided by their wisdom. Chet Bowers (2001) proposes new approaches to teaching that will reshape cultures and foster community, rather than reinforce individualism and separation from the nonhuman natural world. Networks of women across the world have shared ecological wisdom to change the way human beings conceptualize and live in the world (Johnson and Wilkinson 2020; Gebara 1999, 2003; Ruether 2006; Halkes 1989). Further, many environmental leaders have sought to build coalitions based on common concerns for moms, food justice, water or land protection, justice for communities of color, and so forth. These efforts all focus on community.

I have erred on the side of sharing many narratives because practical theologians need the wisdom of many persons, communities, creatures, and habitats if we are to respond to the weeping planet. We can no longer prevent climate crisis; the crisis is here. We *can* still respond to the crisis with a full range of proactive, anticipatory, and reactive responses. All are needed to prevent further destruction and to help regenerate that which can be regenerated. Can you imagine a community of practical theologians who accept responsibility to act with and for the good of our planetary community? We could become

a network of scholars and activists who learn from the wisdom of diverse peoples and forests and deserts, seeking together to halt global warming and heal the earth.

**Funding:** This project received no external funding.

**Institutional Review Board Statement:** Not applicable.

**Informed Consent Statement:** Not applicable.

**Conflicts of Interest:** The author declares no conflict of interest.

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
