# Peer review of "Responding to a Weeping Planet: Practical Theology as a Discipline Called by Crisis"

_religions, doi:10.3390/rel13030244_

Round 1

Reviewer 1 Report

This is a good and important article, well argued, informed by past publications and the most recent debate. It is clear that the author is very well acquainted with the discussion and manages to convincingly argue that the relationship between practice and worldview is critical for the ecological crisis. Her usage of indigenous knowledge systems is not only of value for North America, but also for us in Africa where we can learn much from her approach. My only critical comment pertains to references. She has many excellent references in the paper, however not in the first section where the crises are discussed. Personally I think a few scholarly publications in that section can strengthen this article even more.

Author Response

I am very grateful for your review and will add documentation to the first section. I was drawing on what I found to be broadly shared analyses, which is why I did not document the summary analysis; however, I think your suggestion is a very good one. I am especially relieved to read your comment regarding relevance in African contexts. Thank you for making that connection, and thank you for your careful reading.

Reviewer 2 Report

I enjoyed reading this piece and think it should be published as is. However, I would like to challenge the author in relation to the thrust of the article. All the experts on climate change say we have at most ten years to get the settings correct, to start lowering our CO2 emissions rapidly. does the author really think that the proposals offered will prove effective on that time scale? There is only one mention of "political advocacy" yet it seems to me that the only actions which can address the problem on the timeframe needed is political action, massive and immediate poliicy shifts on CO2 emisisons. The sort of changes the author proposes may take decades to amke a significant difference to our CO2 emissions, by which stage the game will be over. we will face widespread habitat destruction, lose of species, sea level rise and so on. Action is needed which will operate on a decade-long time frame. It will be painful and costly, involving sacrifice of life-styles and facing the violence of vested interests. This is a call to discipleship. what is needed to fill this out is a theory of social and political change, and its application. 

Author Response

Dear Reviewer.

I appreciate your review and concerns about my manuscript, and I plan to review and revise to speak more forthrightly to the urgency of climate change. I do seek to recognize the urgency of our global situation and to propose social and political change, attending to changes in ways of thinking and living as a vital part of political action. I recognize, however, that I am speaking to the crises as habitual and anticipatory as well as catastrophic, thus to actions that are proactive, anticipatory, and reactive. I think I need to underscore the catastrophic situation more than I have and to highlight the urgency of immediate reactive responses. This would not change the focus of the essay but would acknowledge more fully the very important issue that you raise - a concern that I share. Thank you for your thoughtful reading.